# Effective and Robust Detection of Adversarial Examples via Benford-Fourier Coefficients

## Abstract

Adversarial examples have been well known as a serious threat to deep neural networks (DNNs). To ensure successful and safe operations of DNNs on real-world tasks, it is urgent to equip DNNs with effective defense strategies. In this work, we study the detection of adversarial examples, based on the assumption that the output and internal responses of one DNN model for both adversarial and benign examples follow the generalized Gaussian distribution (GGD), but with different parameters (*i.e.*, shape factor, mean, and variance). GGD is a general distribution family to cover many popular distributions (*e.g.*, Laplacian, Gaussian, or uniform). It is more likely to approximate the intrinsic distributions of internal responses than any specific distribution. Besides, since the shape factor is more robust to different databases rather than the other two parameters, we propose to construct discriminative features via the shape factor for adversarial detection, employing the magnitude of Benford-Fourier coefficients (MBF), which can be easily estimated using responses. Finally, a support vector machine is trained as the adversarial detector through leveraging the MBF features. Through the Kolmogorov-Smirnov (KS) test (Massey Jr (1951)), We empirically verify that: 1) the posterior vectors of both adversarial and benign examples follow GGD; 2) the extracted MBF features of adversarial and benign examples follow different distributions. Extensive experiments in terms of image classification demonstrate that the proposed detector is much more effective and robust on detecting adversarial examples of different crafting methods and different sources, in contrast to state-of-the-art adversarial detection methods.

## 1 Introduction

Deep neural networks (DNNs) have achieved a remarkable success in many important applications, such as image classification, face recognition, object detection, *etc*. In the meanwhile, DNNs have been shown to be very vulnerable to adversarial examples. However, many real-world scenarios have very restrictive requirements about the robustness of DNNs, such as face verification for login, or semantic segmentation in autonomous driving. Adversarial examples are a serious threat to the applications of DNNs to these important tasks. Since many kinds of adversarial attack methods have been proposed to fool DNNs, it is more urgent to equip effective defensive strategies to ensure the safety of deep models in real-world applications. However, defense seems to be more challenging than attack, as it has to face adversarial examples from unknown crafting methods and unknown data sources. Typical defensive strategies include adversarial training, adversarial de-noising, and adversarial detection. Compared to the former two strategies, adversarial detection is somewhat more cost-effective, as it often does not need to re-train or modify the original DNN model.

There are two main challenges for adversarial detection. **(1)** The adversarial examples are designed to camouflage themselves to be close to the corresponding benign examples in the input space. Then, where and how to extract the discriminative information to train the detector? **(2)** The data sources and the generating methods of adversarial examples are often inaccessible to the detector. In this case, the detector can be stably effective across different data sources and different attack methods? In other words, a good adversarial detector is required to be not only *effective* to distinguish adversarial and benign examples, but also *robust* to different data sources and attack methods.

To satisfy the first requirement of effectiveness, we utilize the other principle of crafting adversarial examples that the outputs between benign and adversarial examples should be large, to encourage the change of the final prediction. It means that the imperceptible difference between benign and adversarial examples in the input space is enlarged along the DNN model, leading to the significant difference in the output space. Inspired by this fact, we assume that the output or the responses of internal layers of the DNN model should include the discriminative information for benign and adversarial examples. A few works have attempted to extract different types of discriminative features from the output or the internal responses, such as kernel density (KD) (Feinman et al. (2017)) and the local intrinsic dimensionality (LID) (Ma et al. (2018)), *etc*. To achieve the robustness, the extracted discriminative features should model the intrinsic difference between adversarial and benign examples, rather than the difference from the changes of data sources or attack methods. Many existing methods have shown the effectiveness to some extent on detecting adversarial examples of specific data sources and attack methods. However, their robustness, especially across different data sources, has not been well studied and verified.

In this work, we propose a novel detection method based on the assumption that the internal responses of both adversarial and benign examples follow the generalized Gaussian distribution (GGD) (Varanasi & Aazhang (1989)), but with different parameters, including the shape factor, mean, and variance. The rationale behind this assumption is that GGD covers many popular distributions with varied shape factors (such as Laplacian, Gaussian, or uniform distribution), such that GGD is more likely to approximate the intrinsic response distributions rather than one specific distribution. Moreover, mean and variance of GGD may vary significantly with respect to different classes and data sources, even for benign examples, while the shape factor is more robust. For example, the mean and variance of two Gaussian distributions could be totally different, but their shape factors are the same (*i.e.*, 2). Thus, we propose to use the shape factor as an effective and robust discrimination between adversarial and benign examples. However, it is difficult to exactly estimate the shape factor in practice. We resort to the magnitude of Benford-Fourier coefficients (Pasquini et al. (2014)), which is a function of the shape factor. It can be easily estimated using internal responses, according to the definition of Fourier transform. Then, the magnitudes estimated from internal responses of different convolutional layers are concatenated as a novel representation. Finally, a support vector machine (SVM) (Vapnik (2013)) is trained using the new representations as the adversarial detector. Extensive experiments carried out on several databases verify the effectiveness and robustness of the proposed detection method. To further verify the rationale of our assumption, we present the empirical analysis through the Kolmogorov-Smirnov test (KS test) (Massey Jr (1951)). The KS test verifies that 1) the posterior vectors of both adversarial and benign examples predicted by the CNN model follow the distribution of GGD, but with different parameters, and 2) the MBF features of adversarial and benign examples follow different distributions, and the MBF features of adversarial examples crafted from different attack methods follow the same distribution, as well as that the MBF features of adversarial/benign examples from different data sources follow the same distribution. Moreover, we visualize the statistics (*i.e.*, mean ± standard deviation) of the extracted MBF features for adversarial and benign examples. The visualization reveals the distinct difference between adversarial and benign examples. These empirical analysis demonstrates the effectiveness and robustness of the proposed MBF detector.

## 2 RELATED WORK

In this section we focus on reviewing existing works about adversarial detection, while those of other adversarial defensive strategies are out of scope of this work.

The general idea of most existing detection methods is learning or constructing a new representation to discriminate adversarial and benign examples, utilizing the outputs or immediate responses of an original classification network. Li & Li (2017) trained a cascading classifier based on the principal component analysis (PCA) (Pearson (1901)) statistics of responses from each convolutional layer of the defended convolutional neural network (CNN) model. An example is recognized as benign if all single classifiers of the cascade predict it as benign, otherwise adversarial. Lu et al. (2017) proposed SafetyNet by adding a RBF-SVM classifier to detect adversarial examples, at the end of the original classification network. Metzen et al. (2017) proposed a detection network along with an original classification network, which takes the internal responses of the original network as inputs. It shows effectiveness on detecting adversarial examples generated by simple attacks (such as FGSM

(Goodfellow et al. (2014)) and JSMA (Papernot et al. (2016))), while performs much worse when facing more advanced attacks (such as C&W (Carlini & Wagner (2017a))). It tells that this method is sensitive to attack methods. Grosse et al. (2017) attempted to detect adversarial examples using the statistical test of maximum mean discrepancy (MMD). Although above detection methods show effectiveness on some attack methods and some databases, but a thorough evaluation presented in Carlini & Wagner (2017b) has shown that these methods are sensitive to attack methods or databases, and they can be somewhat easily invaded by new attacks.

Some recent works proposed to utilize neighboring samples in the same database to construct a better representation of a current sample. Feinman et al. (2017) defined two metrics based on the responses of the final hidden layer of the classification neural network, including kernel density estimation (KDE) and Bayesian neural network uncertainty (BU). If the metric score of KDE/BU is lower/higher than a pre-defined threshold, then the example is predicted as adversarial. Ma et al. (2018) utilized the local intrinsic dimensionality (LID) to measure the characterization of adversarial regions of DNNs. LID describes the distance between one example and its $k$-nearest neighboring sample in the feature space of immediate responses of the original classification network. The distances computed from different layers are concatenated as the example representation, which is then used to train a shallow classifier to discriminate adversarial and benign examples. Zheng & Hong (2018) defined the intrinsic hidden state distribution (IHSD) of the responses of the original classification network to model different classes. The Gaussian mixture model (GMM) was used to approximate IHSD of each class. Then, the posterior probability of one sample assigned to GMM is computed as the metric. If the probability is lower than a pre-defined threshold, then it is recognized as adversarial. Lee et al. (2018) computed the class-conditional Gaussian distribution of the responses of the original classification network based on the whole training set. Then, the Mahalanobis distance between one sample and its nearest class-conditional Gaussian distribution is used as the metric for detection. If the distance is larger than a pre-defined threshold, then it is detected as adversarial. Compared to some aforementioned single-representation-based detection methods, these joint-representation-based methods showed better performances on some databases. However, the detection cost for each example is much higher, as the responses of its neighboring samples should also be computed. Besides, since the representation is highly dependent on the neighbors or all training examples, the detection performance may be sensitive to data sources, which will be studied in later experiments.

There are also some other approaches that do not construct representations from the responses of the original classification network. Hendrycks & Gimpel (2016) adopted PCA statistics to discriminate adversarial and benign images, independent of any DNN model. However, the study presented in Carlini & Wagner (2017b) has demonstrated that this method works for MNIST but not for CIFAR-10, and PCA statistics are not robust features to detect adversarial images. Pang et al. (2018) proposed a novel loss called reverse cross-entropy (RCE) to train the classification network, such that the distance measured by kernel density (Feinman et al. (2017)) between adversarial and benign examples could be enlarged. Samangouei et al. (2018) proposed Defense-GAN to model the distribution of benign examples using a generative adversarial network (GAN). If the Wasserstein distance between one example and its corresponding example generated by the GAN is larger than a pre-defined value, then it is detected as adversarial. However, these above three methods are much more costly than other detection methods.

## 3 PRELIMINARIES

### 3.1 GENERALIZED GAUSSIAN DISTRIBUTION

Assume that a random variable $\mathcal{X} \in \mathbb{R}^d$ follows the generalized Gaussian distribution (GGD) (Varanasi & Aazhang (1989)). Then, its probability density function (PDF) is formulated with two positive parameters, including the shape factor $c$ and the standard deviation $\sigma$, as follows

$$\mathcal{P}_{\mathcal{X}}(x) = A \cdot e^{-|\beta x|^c}, \tag{1}$$

where $\beta = \frac{1}{\sigma}\left(\frac{\Gamma(3/c)}{\Gamma(1/c)}\right)^{\frac{1}{2}}$ and $A = \frac{\beta c}{2\Gamma(1/c)}$, with $\Gamma(\cdot)$ being the Gamma function. Note that the mean parameter $\mu$ is omitted above, as $\mu$ has no relation with the shape of distribution and we set it as $0$ without loss of generality. A nice characteristic of GGD is that it covers many popular distributions

with varied shape factors. For example, when $c = 1$, then it becomes the Laplacian distribution; when $c = 2$, then it is the Gaussian distribution with a variance of $\sigma^2$; when $c \to +\infty$, then it is specified as a uniform distribution on $(-\sqrt{2}\sigma, \sqrt{2}\sigma)$.

## 3.2 BENFORD-FOURIER COEFFICIENTS

Although generalized Gaussian distribution (GGD) is able to cover a bunch of distributions, it is hard to depict the exact forms of GGD precisely. To this end, we further define a random variable $\mathcal{Z} = \log_{10} |\mathcal{X}| \mod 1$ for detecting and distinguishing different form of GGD, of which the PDF is formulated by means of Fourier Series as Pérez-González et al. (2007), with the fundamental period being fixed as $2\pi$,

$$\mathcal{P}_{\mathcal{Z}}(z) = 1 + 2 \sum_{n=1}^{+\infty} [A_n \cos(2\pi nz) + B_n \sin(2\pi nz)] = 1 + 2 \sum_{n=1}^{+\infty} |a_n| \cos(2\pi nz + \phi_n), \quad (2)$$

where $z \in [0, 1)$ corresponds to the domain of random variable $\mathcal{Z}$, the phase of Fourier Series is explained as $\phi_n = \arctan\left(-\frac{B_n}{A_n}\right)$, and the magnitude denotes $|a_n| = \sqrt{A_n^2 + B_n^2}$. $a_n = |a_n| \cdot e^{j\phi_n}$ denotes the $n$-th Fourier coefficient of $\mathcal{P}_{\mathcal{Z}}(z)$ evaluated at $2\pi n$, and its definition is

$$a_n = \int_{-\infty}^{+\infty} \mathcal{P}_{\mathcal{Z}}(z) \cdot e^{-j2\pi n \log_{10} z} dz = \frac{2A e^{\frac{j2\pi n \log \beta}{\log 10}}}{\beta c} \cdot \Gamma\left(\frac{-j2\pi n + \log 10}{c \log 10}\right). \quad (3)$$

$a_n$ is also called as Benford-Fourier coefficient. Note that $a_n$ is a complex number, and its magnitude can be calculated as follows

$$|a_n| = \left(\prod_{k=0}^{+\infty} \left[1 + \left(\frac{2\pi n}{\log 10(ck + 1)}\right)^2\right]^{-1}\right)^{\frac{1}{2}}. \quad (4)$$

Note that $|a_n|$ gets smaller as $n \in \mathbb{N}$ increases. And, an interesting property of $|a_n|$ is that it only depends on the shape factor $c$, while is independent of the parameter $\sigma$. Thus, one set of the absolute values of Benford-Fourier coefficients $\{|a_n|\}_{n \in \mathbb{N}}$ correspond to one identical $c$, *i.e.*, one identical special distribution of GGD. In other words, we could use $\{|a_n|\}_{n \in \mathbb{N}}$ as features or representations to discriminate different special distributions of GGD.

However, if it is often difficult to know or even estimate the shape factor $c$, we cannot compute the value of $|a_n|$. But fortunately, recalling that $a_n$ is the $n$-th Fourier coefficient of $\mathcal{P}_{\mathcal{Z}}(z)$ evaluated at $2\pi n$, we can derive an easy estimation. Specifically, assume that $\mathbf{x} = \{x_1, \ldots, x_M\}$ is a set of $M$ i.i.d. points sampled from GGD with the same shape factor $c$. Then, the corresponding Benford-Fourier coefficients can be estimated as follows (Pasquini et al. (2014)):

$$\hat{a}_n = \frac{\sum_{m=1}^{M} e^{-j2\pi n \log_{10} |x_m|}}{M} = \frac{1}{M} \sum_{m=1}^{M} \left[\cos(2\pi n \log_{10} |x_m|) - j \sin(2\pi n \log_{10} |x_m|)\right]. \quad (5)$$

The gap between $\hat{a}_n$ and $a_n$ is analyzed in Theorem 1. It tells that $\hat{a}_n$ gets closer to $a_n$ as $M$ increases. For clarity, we firstly introduce a few notations: $\mathcal{T} = e^{-j2\pi n \log_{10} |\mathcal{X}|}$ is a random variable with $\mathcal{X}$ obeying the generalized Gaussian distribution, and $\hat{a}_n$ is an observation of the random variable $\mathcal{Y} = \frac{1}{M} \sum_{m=1}^{M} \mathcal{T}_m$. Due to the space limit, the proof will be presented in **supplementary material**.

**Theorem 1** *Assume that the estimation error $\varepsilon_n = \hat{a}_n - a_n$ is an observation of the random variable $\mathcal{E} = \mathcal{Y} - a_n$. $|\mathcal{E}|$ follows the Rayleigh distribution (Papoulis & Pillai (2001)), of which the probability density function (PDF) is formulated as*

$$\mathcal{P}_{|\mathcal{E}|}(r) = 2Mre^{-Mr^2}. \quad (6)$$

*And the expectation and variance are respectively shown as follows:*

$$E(|\mathcal{E}|) = \frac{1}{2}\sqrt{\frac{\pi}{M}}, D(|\mathcal{E}|) = \frac{4 - \pi}{4M},$$

*which implies that the estimation error $\varepsilon_n$ gets closer to 0 as the number of samples $M$ increases.*

# 4 ADVERSARIAL DETECTION VIA BENFORD-FOURIER COEFFICIENTS

## 4.1 TRAINING PROCEDURE OF ADVERSARIAL DETECTOR

There are three stages to train the proposed adversarial detector, including: **1)** building a training set based on benign images; **2)** extracting novel representations of the training set via Benford-Fourier coefficients; **3)** training a SVM classifier as the adversarial detector. They will be explained in details sequentially. And, the overall training procedure is briefly summarized in Algorithm 1.

**Build a training set**. Firstly, we collect $N$ clean images $\{\mathbf{x}_1, \ldots, \mathbf{x}_N\}$, which can be correctly predicted by $f_{\boldsymbol{\theta}}$. Then, we adopt one adversarial attack method (*e.g.*, C&W (Carlini & Wagner (2017a)) or BIM ( Kurakin et al. (2016))) to generate one adversarial image corresponding to each clean image. The crafted $N$ adversarial examples are denoted as $\{\hat{\mathbf{x}}_1, \ldots, \hat{\mathbf{x}}_N\}$. Besides, to avoid that the noisy image (polluted by some kind of non-malicious noises but still can be correctly predicted by $f_{\boldsymbol{\theta}}$) is incorrectly detected as adversarial, we also craft one noisy image by adding small random Gaussian noises onto each clean image. These $N$ noisy examples are denoted as $\{\bar{\mathbf{x}}_1, \ldots, \bar{\mathbf{x}}_N\}$. Note that, hereafter benign examples include both clean and Gaussian noisy examples. Consequently, we obtain one training set with $3N$ examples, denoted as $\mathcal{D}_{tr} = \{(\mathbf{x}_i, -1), (\bar{\mathbf{x}}_i, -1), (\hat{\mathbf{x}}_i, +1)\}_{i=1,\ldots,N}$.

**Extract novel representations**. We firstly feed the $i$-th training image from $\mathcal{D}_{tr}$ into $f_{\boldsymbol{\theta}}$. We concatenate all response entries of the $l$-th layer in $f_{\boldsymbol{\theta}}$ to obtain one vector $\mathbf{r}_i^l$. Then, we estimate the corresponding Benford-Fourier coefficients according to Eq. (5), as follows

$$(\hat{a}_n)_i^l = \frac{1}{M_i^l} \sum_{m=1}^{M_i^l} e^{-j2\pi n \log_{10} |(r_m)_i^l|}, \tag{7}$$

where $M_i^l$ indicates the length of $\mathbf{r}_i^l$. The magnitude of $(\hat{a}_n)_i^l$ is computed as follows

$$|(\hat{a}_n)_i^l| = \frac{1}{M_i^l} \sum_{m=1}^{M_i^l} \left( \cos^2 \left(2\pi n \log_{10} |(r_m)_i^l|\right) + \sin^2 \left(2\pi n \log_{10} |(r_m)_i^l|\right) \right)^{\frac{1}{2}}. \tag{8}$$

Then, we extract one $T$-dimensional feature vector $\mathbf{a}_i^l = [|(\hat{a}_1)_i^l|, \ldots, |(\hat{a}_T)_i^l|] \in \mathbb{R}_+^T$ for the $i$-th training image from the $l$-th layer. We set $T = 16$ in experiments, as $|(\hat{a}_n)_i^l|$ of larger $n$ is too small for discrimination. Finally, we concatenate the feature vectors of all layers to form a long vector $\hat{\mathbf{a}}_i = [\hat{\mathbf{a}}_i^1; \ldots; \hat{\mathbf{a}}_i^L] \in \mathbb{R}_+^{TL}$, with $L$ being the number of layers in $f_{\boldsymbol{\theta}}$. Consequently, we obtain a novel representation of the training set $\mathcal{D}_{tr}$, denoted as $\hat{\mathcal{A}}_{tr} = \{(\hat{\mathbf{a}}_i, \pm 1)\}_{i=1,\ldots,3N}$, where the label of $\mathbf{a}_i$ (*i.e.*, $+1$ or $-1$) is directly obtained from $\mathcal{D}_{tr}$.

**Train an adversarial detector**. Finally, we train a binary SVM classifier based on $\hat{\mathcal{A}}_{tr}$. The trained SVM classifier will serve as the adversarial detector for the CNN model $f_{\boldsymbol{\theta}}$.

**Testing**. One novel testing example is firstly predicted as adversarial or not by the trained adversarial detector . If adversarial, then it is rejected; otherwise, it is fed into $f_{\boldsymbol{\theta}}$ to predict its class label.

**Remark**. Note that in the derivation of $a_n$ (see Eq. (3)), the mean parameter of GGD is set to 0. In experiments, we calculate the mean values of internal-layer responses of all networks for every image, and find that mean parameters at most layers are close to 0, while the mean parameters at a few layers could be large. However, the mean value is subtracted from each response entry when we extract the MBF features in our experiments. Thus, the derived $a_n$ is applicable to our task. Besides, according to Theorem 1, the estimation error of $(\hat{a}_n)_i^l$ is inversely proportional to $M_i^l$. It tells that the coefficient estimated from the larger-sized layer is more accurate. In many neural networks (*e.g.*, AlexNet (Krizhevsky et al. (2012))), the response sizes of high layers get smaller, which means the less accurate estimation. However, we believe that the discrimination between adversarial and benign examples in higher layers is more evident than that in lower layers. There is a trade-off between estimation accuracy and discrimination. This is why we concatenate the estimated magnitudes of all layers together to construct the novel representation.

## 4.2 EXPERIMENTAL SETTINGS

**Databases and network architectures** We conduct experiments on three databases, including CIFAR-10 (Krizhevsky et al. (2014)), SVHN (Netzer et al. (2011)), and a subset of ImageNet (Deng

---

**Algorithm 1** Training the adversarial detector via the magnitude of Benford-Fourier coefficients.

---

**Require:** The trained CNN model $f_{\boldsymbol{\theta}}$ with $L$ layers, and the training set $\mathcal{D}_{tr}$
1: **for** $i = 1$ to $|\mathcal{D}_{tr}|$ **do**:
2:     **for** $l = 1$ to $L$ **do**:
3:         Compute $|(\hat{a}_n)_i^l|$ as Eq. (8), with $n = 1, \ldots, T$ ;
4:         Concatenate $\{|(\hat{a}_n)_i^l|\}_{n=1,\ldots,T}$ to obtain a vector $\hat{\mathbf{a}}_i^l$ ;
5:     **end for**
6:     Concatenate $\{\hat{\mathbf{a}}_i^l\}_{l=1,\ldots,L}$ to obtain a long vector $\hat{\mathbf{a}}_i$ ;
7: **end for**
8: Build a novel representation of the training set, denoted as $\hat{\mathcal{A}}_{tr} = \{(\hat{\mathbf{a}}_i, \pm 1)\}_{i=1,\ldots,|\mathcal{D}_{tr}|}$ ;
9: Train a binary SVM classifier based on $\hat{\mathcal{A}}_{tr}$ ;
    **return** The trained binary SVM classifier.

---

et al. (2009)). In terms of CIFAR-10 and SVHN, we adopt the same settings as the compared method LID (Ma et al. (2018)). Specifically, a 33-layer network pre-trained on the training set of CIFAR-10 achieves $82.37\%$ accuracy on the testing set with $10,000$ benign images; a 19-layer network pre-trained on the training set of SVHN achieves $92.6\%$ accuracy on the testing set with $26,032$ benign images. Then, we add a small noise drawn from $\mathcal{N}(0, \sigma^2)$ on each testing image, with $\sigma$ being the similar level of the $\ell_2$ norm of adversarial perturbations on the same database. If both the benign and its noisy image can be correctly predicted by the classification network, then it is picked out for training the detection. We finally collect $8,175$ and $23,862$ benign images from CIFATR-10 and SVHN, respectively. These images are randomly partitioned to the $80\%$ training set and the $20\%$ testing set, used for the training and testing of the detector. We also collect a subset from ImageNet, including 800 benign images of 8 classes (*snowbird, spoonbill, bobtail, Leonberg, hamster, proboscis monkey, cypripedium calceolus, and earthstar*). The 100 images of each class contain 50 testing images and 50 randomly selected training images. We fine-tune the checkpoints of both AlexNet and VGG pre-trained on ImageNet[1] on these 800 images to achieve $100\%$ accuracy. Then, 785 benign images are kept for detection, as both their noisy images and themselves can be correctly predicted by both the fine-tuned AlexNet and VGG models. These 785 images are then randomly partitioned to 400 training and 305 testing images used for detection. For each database, as described in Section 4.1, one noisy and one adversarial image are generated for each benign image; then, all of benign, noisy, and adversarial images are used for detection.

**Attack methods** We adopt four popular adversarial attack methods to craft adversarial examples, including basic iterative method (BIM (Kurakin et al. (2016))), CarliniWagnerL2Attack (CW-L2 (Carlini & Wagner (2017a))), DeepFool ( (Moosavi-Dezfooli et al. (2016))), and random projected gradient descent (R-PGD (Madry et al. (2017))). They are implemented by Foolbox[2].

**Compared detection methods** We compare with three state-of-the-art and open-sourced adversarial detection methods, including KD+BU[3] (Feinman et al. (2017)), Mahalanobis distance[4] (M-D) (Lee et al. (2018)), and LID[5] (Ma et al. (2018)). Note that another recent work called I-defender (Zheng & Hong (2018)) is not compared, as its code is not available. To ensure the fair comparison, the SVM classifier is trained with all compared methods, implemented by the *fitcsvm*[6] function in MAT-LAB. There are two important hyper-parameters in LID, *i.e.*, the size of mini-batch and the number of neighbors. On CIFAR-10 and SVHN, they are respectively set as 100 and 20, as suggested in Ma et al. (2018); on ImageNet, as there are only 400 benign training images, they are respectively set as 50 and 20 in experiments. Moreover, we find that there are some unfair settings in the implementations of compared methods. For example, KD+BU utilizes the extra $50,000$ images of CIFAR-10 to compute the kernel density of each training and testing image; M-D also uses these extra images to compute the mean and co-variance of GMM. Since extra images of the similar distribution with the training images are often unavailable, we believe that extra images should not be used to ensure the fair comparison. Thus, extra images are not used for KD+BU and M-D in our experiments. Besides,

---

[1]https://pytorch.org/docs/robust/torchvision/models.html

[2]https://foolbox.readthedocs.io/en/latest/

[3] https://github.com/rfeinman/detecting-adversarial-samples/

[4] https://github.com/pokaxpoka/deep_Mahalanobis_detector/

[5] https://github.com/xingjunm/lid_adversarial_subspace_detection/

[6]https://www.mathworks.com/help/stats/fitcsvm.html

Table 1: Detection results in the *non-transfer* case.

| database | Detector | AUROC (%) | | | | Accuracy (%) | | | |
|---|---|---|---|---|---|---|---|---|---|
| | | BIM | CW-L2 | DeepFool | R-PGD | BIM | CW-L2 | DeepFool | R-PGD |
| CIFAR-10 | KD+BU (Feinman et al. (2017)) | 79.0 | 82.8 | 80.6 | 77.7 | 74.2 | 73.0 | 71.7 | 72.6 |
| | M-D (Lee et al. (2018)) | 51.7 | 48.4 | 53.7 | 52.0 | 66.7 | 66.7 | 66.7 | 66.7 |
| | LID (Ma et al. (2018)) | 87.9 | 87.4 | 86.6 | 82.7 | 72.6 | 78.5 | 78.0 | 70.2 |
| | MBF | **99.6** | **99.6** | **96.9** | **99.4** | **98.8** | **98.3** | **91.8** | **98.5** |
| SVHN | KD+BU (Feinman et al. (2017)) | 81.5 | 85.1 | 84.1 | 82.5 | 78.6 | 80.0 | 79.3 | 78.6 |
| | M-D (Lee et al. (2018)) | 49.7 | 50.0 | 49.6 | 50.1 | 66.7 | 66.7 | 66.7 | 66.7 |
| | LID (Ma et al. (2018)) | 91.7 | 88.7 | 92.2 | 90.5 | 83.3 | 80.8 | 84.0 | 81.5 |
| | MBF | **99.7** | **99.9** | **99.3** | **99.5** | **98.8** | **98.3** | **91.8** | **98.5** |
| ImageNet-AlexNet | KD+BU (Feinman et al. (2017)) | 50.8 | 52.6 | 51.1 | 51.5 | 34.4 | 36.8 | 34.9 | 35.3 |
| | M-D (Lee et al. (2018)) | 48.1 | 60.1 | 46.5 | 54.1 | 66.7 | 66.6 | 66.8 | 66.7 |
| | LID (Ma et al. (2018)) | 71.7 | 70.9 | 71.9 | 72.3 | 68.7 | 60.5 | 65.8 | 68.3 |
| | MBF | **99.9** | **99.6** | **99.9** | **99.8** | **98.6** | **97.8** | **98.8** | **98.4** |
| ImageNet-VGG16 | KD+BU (Feinman et al. (2017)) | 57.1 | 57.7 | 56.2 | 58.6 | 42.8 | 43.6 | 41.7 | 44.8 |
| | M-D (Lee et al. (2018)) | 63.7 | 64.8 | 45.9 | 66.9 | 67.1 | 65.6 | 66.7 | 67.4 |
| | LID (Ma et al. (2018)) | 82.6 | 84.2 | 89.5 | 84.2 | 77.8 | 76.7 | 83.3 | 76.4 |
| | MBF | **99.8** | **100.0** | **100.0** | **100.0** | **99.6** | **99.6** | **100.0** | **100.0** |

Table 2: Detection results evaluated by AUROC (%) in the *attack-transfer* case.

| database | Test attack → | BIM | CW-L2 | DeepFool | R-PGD |
|---|---|---|---|---|---|
| | Train attack ↓ | KD+BU (Feinman et al. (2017)) / M-D (Lee et al. (2018)) / LID (Ma et al. (2018)) / MBF | | | |
| CIFAR-10 | BIM | 79.0 / 51.7 / 87.9 / **99.6** | 78.1 / 52.2 / 85.7 / **99.4** | 76.4 / 52.4 / 86.4 / **87.8** | 78.6 / 50.9 / 87.6 / **99.4** |
| | CW-L2 | 83.4 / 47.9 / 90.3 / **99.6** | 82.8 / 48.4 / 87.4 / **99.6** | 80.3 / 46.5 / 87.0 / **88.6** | 82.9 / 48.2 / 88.9 / **99.5** |
| | DeepFool | 83.6 / 53.5 / 89.7 / **99.3** | 83.1 / 51.4 / 86.1 / **99.1** | 80.6 / 53.7 / 86.6 / **96.9** | 83.1 / 53.6 / 87.9 / **98.9** |
| | R-PGD | 78.1 / 51.5 / 86.1 / **99.6** | 77.3 / 51.9 / 84.5 / **99.5** | 75.8 / 52.4 / 84.9 / **88.3** | 77.7 / 52.1 / 82.7 / **99.4** |
| SVHN | BIM | 81.5 / 49.7 / 91.7 / **99.7** | 83.1 / 51.0 / 88.3 / **99.7** | 80.1 / 48.8 / 91.9 / **97.5** | 81.5 / 50.4 / 90.7 / **99.5** |
| | CW-L2 | 83.6 / 50.8 / 91.6 / **99.7** | 85.1 / 50.0 / 88.7 / **99.9** | 82.5 / 49.7 / 92.0 / **97.3** | 83.6 / 49.1 / 90.8 / **99.7** |
| | DeepFool | 85.0 / 50.0 / 91.6 / **99.7** | 86.3 / 50.0 / 87.9 / **99.8** | 84.1 / 49.6 / 92.2 / **99.3** | 85.0 / 49.4 / 90.8 / **99.6** |
| | R-PGD | 82.5 / 49.9 / 91.2 / **99.6** | 84.0 / 49.5 / 87.8 / **99.6** | 81.2 / 50.2 / 91.6 / **97.4** | 82.5 / 50.1 / 90.5 / **99.5** |
| ImageNet-AlexNet | BIM | 50.8 / 48.1 / 71.7 / **99.9** | 50.8 / 44.9 / 70.8 / **99.6** | 50.4 / 49.7 / 72.1 / **99.8** | 50.7 / 50.4 / 70.8 / **99.8** |
| | CW-L2 | 52.3 / 50.7 / 69.7 / **99.8** | 52.6 / 60.1 / 70.9 / **99.7** | 51.2 / 52.1 / 66.0 / **99.7** | 52.2 / 55.8 / 69.8 / **99.8** |
| | DeepFool | 51.8 / 53.6 / 71.3 / **99.9** | 51.9 / 53.5 / 69.7 / **99.6** | 51.2 / 46.5 / 71.9 / **99.9** | 51.8 / 51.2 / 71.2 / **99.8** |
| | R-PGD | 51.6 / 52.4 / 72.5 / **99.8** | 51.3 / 50.4 / 72.3 / **99.6** | 50.3 / 50.5 / 72.2 / **99.7** | 51.5 / 54.1 / 72.3 / **99.8** |
| ImageNet-VGG-16 | BIM | 57.1 / 63.7 / 82.6 / **99.8** | 57.1 / 57.2 / 84.3 / **99.5** | 53.8 / 54.4 / 90.4 / **100.0** | 57.1 / 65.4 / 84.2 / **100.0** |
| | CW-L2 | 57.3 / 64.3 / 81.9 / **100.0** | 57.7 / 64.8 / 84.2 / **100.0** | 53.8 / 56.6 / 90.2 / **100.0** | 57.5 / 67.6 / 84.3 / **100.0** |
| | DeepFool | 58.7 / 45.5 / 80.2 / **99.7** | 59.4 / 46.1 / 79.1 / **99.3** | 56.2 / 45.9 / 89.5 / **100.0** | 59.0 / 44.0 / 80.9 / **99.7** |
| | R-PGD | 58.6 / 62.3 / 81.1 / **100.0** | 58.9 / 60.8 / 83.7 / **100.0** | 55.3 / 54.4 / 90.8 / **100.0** | 58.6 / 66.9 / 84.2 / **100.0** |

Table 3: Detection results evaluated by accuracy (%) in the *attack-transfer* case.

| database | Test attack → | BIM | CW-L2 | DeepFool | R-PGD |
|---|---|---|---|---|---|
| | Train attack ↓ | KD+BU (Feinman et al. (2017)) / M-D (Lee et al. (2018)) / LID (Ma et al. (2018)) / MBF | | | |
| CIFAR-10 | BIM | 74.2 / 66.7 / 72.6 / **98.8** | 73.0 / 66.7 / 66.9 / **97.4** | 71.7 / 66.7 / 70.2 / **87.3** | 73.8 / 66.7 / 61.3 / **96.8** |
| | CW-L2 | 74.2 / 66.7 / 80.3 / **98.1** | 73.0 / 66.7 / 78.5 / **98.3** | 71.7 / 66.7 / 77.9 / **89.5** | 73.8 / 66.7 / 73.6 / **98.2** |
| | DeepFool | 74.0 / 66.7 / 80.4 / **95.2** | 72.8 / 66.7 / 77.4 / **95.2** | 71.7 / 66.7 / 78.0 / **91.8** | 73.4 / 66.7 / 74.8 / **95.4** |
| | R-PGD | 73.1 / 66.7 / 77.7 / **98.6** | 71.9 / 66.7 / 75.1 / **98.5** | 71.0 / 66.7 / 75.4 / **89.7** | 72.6 / 66.7 / 70.2 / **98.5** |
| SVHN | BIM | 78.6 / 66.7 / 83.3 / **98.7** | 80.5 / 66.7 / 79.4 / **98.9** | 77.7 / 66.7 / 83.5 / **95.8** | 78.6 / 66.7 / 82.9 / **98.2** |
| | CW-L2 | 78.2 / 66.7 / 83.5 / **98.0** | 80.0 / 66.7 / 80.8 / **99.2** | 77.1 / 66.7 / 84.0 / **95.9** | 78.1 / 66.7 / 80.7 / **97.3** |
| | DeepFool | 79.8 / 66.7 / 83.9 / **98.0** | 81.8 / 66.7 / 79.9 / **98.4** | 79.3 / 66.7 / 84.0 / **97.3** | 79.9 / 66.7 / 80.7 / **98.0** |
| | R-PGD | 78.6 / 66.7 / 83.6 / **98.6** | 80.5 / 66.7 / 79.9 / **98.9** | 77.7 / 66.7 / 83.8 / **96.5** | 78.6 / 66.7 / 81.5 / **98.8** |
| ImageNet-AlexNet | BIM | 34.4 / 66.7 / 68.7 / **98.6** | 34.4 / 66.6 / 68.6 / **97.3** | 34.1 / 66.7 / 66.3 / **98.5** | 34.3 / 66.7 / 65.0 / **98.4** |
| | CW-L2 | 36.5 / 66.2 / 64.6 / **98.0** | 36.8 / 66.6 / 60.6 / **97.8** | 35.0 / 66.1 / 60.7 / **98.1** | 36.4 / 66.1 / 62.6 / **98.0** |
| | DeepFool | 36.0 / 66.7 / 69.4 / **98.8** | 36.0 / 66.8 / 65.8 / **97.9** | 34.9 / 66.8 / 65.8 / **98.8** | 35.9 / 66.6 / 68.3 / **98.6** |
| | R-PGD | 35.4 / 66.7 / 67.5 / **98.5** | 35.2 / 66.7 / 67.7 / **97.6** | 34.1 / 66.7 / 67.0 / **98.3** | 34.1 / 66.7 / 68.3 / **98.4** |
| ImageNet-VGG-16 | BIM | 42.8 / 67.1 / 77.8 / **98.6** | 42.8 / 66.8 / 74.5 / **97.3** | 38.4 / 66.0 / 82.7 / **98.5** | 42.7 / 67.9 / 76.9 / **98.4** |
| | CW-L2 | 43.1 / 64.8 / 75.2 / **98.0** | 43.6 / 65.6 / 76.7 / **97.8** | 38.4 / 62.5 / 82.5 / **98.1** | 43.3 / 67.2 / 73.9 / **98.0** |
| | DeepFool | 45.4 / 66.7 / 76.2 / **98.8** | 46.1 / 66.7 / 72.9 / **97.9** | 41.8 / 66.7 / 83.3 / **98.8** | 45.8 / 66.7 / 74.4 / **98.6** |
| | R-PGD | 44.8 / 66.3 / 76.3 / **98.5** | 45.4 / 66.6 / 74.1 / **97.6** | 40.6 / 64.9 / 82.4 / **98.3** | 45.8 / 67.4 / 76.4 / **98.4** |

LID utilizes other benign testing images as neighborhoods to extract features for each testing image. It is unfair to utilize the information of benign or adversarial for neighboring testing images. In our experiments, we use benign training images as neighborhoods for each testing image.

**Three comparison cases and evaluation metrics** We conduct experiments of three cases, including: **1)** *non-transfer*, both training and testing adversarial examples are crafted by the same attack method; **2)** *attack-transfer*, both training and testing adversarial examples are crafted by different attack methods; **3)** *data-transfer*, both training and testing adversarial examples are crafted by the same attack method, but the data sources of training and testing benign examples are different. Two widely used metrics are used to evaluate the detection performance, including area under the receiver operating characteristics (AUROC), and detection accuracy, which is the diagonal summation of the confusion matrix, using $0.5$ as the threshold of the posterior probability. Higher values of both metrics indicate better performance.

## 4.3 RESULTS

Detection results in the *non-transfer* case are shown in Table 1. The proposed MBF method shows the best performance in all cases, and is much superior to all compared methods. LID performs the second-best in most case. KD+BU gives somewhat good detection performance on CIFAR-10 and SVHN, but performs very poor on ImageNet. It tells that KD+BU is very sensitive to different databases and networks. M-D gives almost the degenerate results. Note that the results of M-D on

Table 4: Detection results evaluated by AUROC score (%) in the *data-transfer* case. All detectors are trained on the train set, and tested on the out-of-sample set (including 365 images) of the ImageNet database. The best results are highlighted in bold.

| | ImageNet-AlexNet | | | | ImageNet-VGG-16 | | | |
|---|---|---|---|---|---|---|---|---|
| Detector | BIM | CW-L2 | DeepFool | R-PGD | BIM | CW-L2 | DeepFool | R-PGD |
| KD+BU | 66.3 | 69.4 | 64.7 | 68.8 | 84.0 | 85.0 | 79.9 | 85.2 |
| M-D | 48.7 | 49.2 | 50.0 | 51.0 | 54.5 | 50.0 | 50.4 | 56.1 |
| LID | 69.1 | 67.6 | 72.9 | 71.4 | 84.2 | 88.4 | 89.2 | 83.7 |
| MBF | **99.1** | **99.0** | **99.6** | **99.4** | **99.3** | **99.5** | **99.3** | **99.5** |

Table 5: Attack Failure Rates (%) of adaptive attacks with different detectors. See context for details.

| | Original CW-L2 | LID | MBF |
|---|---|---|---|
| CIFAR-10 | 0.0 | 88.4 | **100.0** |
| SVHN | 0.0 | 85.6 | **92.9** |

CIFAR-10 and SVHN reported in (Lee et al. (2018)) are very high, but their networks (*i.e.*, ResNet (He et al. (2016)) and DenseNet (Huang et al. (2017)) are different from that used in our experiments. It implies that M-D may be suitable for very deep neural networks, but not for shallower networks.

Detection results in the *attack-transfer* case evaluated by AUROC and accuracy are presented in Tables 2 and 3, respectively. MBF still shows much better performance in all transfer cases than all compared methods, and the changes among detecting different attacks are very small. It verifies the robustness of MBF to different attacks.

We also conduct a *data-transfer* experiment on ImageNet. Specifically, we collect extra 365 images of 8 classes (same with the classes used for the detection training, see Section 4.2) through searching the class names in Baidu and Facebook. These 365 benign and their noisy images can be correctly predicted by both the fine-tuned AlexNet and VGG models. The results on these images are shown in Table 4. MBF still shows the best performance. And, compared to the corresponding results in Table 1, the AUROC/accuracy scores of MBF on detecting different attacks change very gently. It verifies the robustness of MBF to different data sources.

### 4.4 Adaptive Attack Against Adversarial Detection

Similar to Carlini & Wagner (2017b) and Ma et al. (2018), we also present an adaptive white-box attack, of which the goal is not only to fool the classifier, but also to evade the adversarial detector. It measures how easy to bypass the detector. Specifically, by combining the CW-L2 attack (Carlini & Wagner (2017a)) and the proposed MBF detector, the adaptive attack is formulated as follows:

$$\arg\min_{\mathbf{x}_{adv} \in [0,1]} \|\mathbf{x} - \mathbf{x}_{adv}\|_2^2 + \alpha \cdot \big[ -l(f_{\boldsymbol{\theta}}(\mathbf{x}_{adv}), y) + \|D(f_{\boldsymbol{\theta}}(\mathbf{x}_{adv})) - D(f_{\boldsymbol{\theta}}(\mathbf{x}))\|_1 \big], \quad (9)$$

where $\mathbf{x}$ denotes the clean input, $y$ indicates the ground-truth label. $l(\mathbf{x}_{adv}, y)$ is the Hinge loss. $D(f_{\boldsymbol{\theta}}(\mathbf{x}_{adv}))$ represents the extracted features by the the detector $D$ (specified later), from $f_{\boldsymbol{\theta}}(\mathbf{x}_{adv})$. The minimization of the $\ell_1$ term $\|D(f_{\boldsymbol{\theta}}(\mathbf{x}_{adv})) - D(f_{\boldsymbol{\theta}}(\mathbf{x}))\|_1$ encourages the detection features of adversarial and benign examples to be close, such that the detector could be evaded.

We conduct the above adaptive attack on both CIFAR-10 and SVHN. Specifically, 1000 clean examples are randomly selected from the test set of each database. For each generated adversarial example, if it fails to fool the classifier $f_{\boldsymbol{\theta}}$, we record 1. The attack failure rate over all 1000 adversarial examples is computed as the metric. The higher rate indicates that it is more difficult to evade the detector. Since LID performs much better than KD+BU and M-D in above experiments, here we only compare LID and the proposed MBF detector. For clarity, the detection features are only extracted from the soft-max layer for both LID and MBF. The original CW-L2 attack (*i.e.*, without the $\ell_1$ term) is also presented as the baseline. With the same setting in (Ma et al. (2018)), the trade-off parameter $\alpha$ is determined by binary search within the range $[10^{-3}, 10^6]$. The results are shown in Table 5. The attack failure rates of MBF on both databases are much higher than those of LID. It tells that it is more difficult to evade MBF than LID.

### 4.5 Hypothesis Test via Kolmogorov-Smirnov Test

**Kolmogorov-Smirnov test** (KS test) (Massey Jr (1951)) is a non-parametric test method in statistics, to test whether a sample follows a reference probability distribution (one-sample KS test), or

Table 6: The $p$-value of KS hypothesis test among clean samples, noisy samples, and adversarial samples crafted by four methods, respectively, on ImageNet-AlexNet.

| ImageNet-VGG-16 | clean | noisy | BIM | CW-L2 | DeepFool | R-PGD |
|---|---|---|---|---|---|---|
| train set | 0.114 | 0.114 | 0.236 | 0.228 | 0.240 | 0.235 |
| test set | 0.114 | 0.114 | 0.231 | 0.228 | 0.239 | 0.232 |

Table 7: The $p$-value of two-sample KS test among MBF coefficients of different types of examples.

| ImageNet-VGG-16 | (clean, noisy) | (clean, BIM) | (BIM, DeepFool) | (CW-L2, R-PGD) |
|---|---|---|---|---|
| train set | 0.998 | 0.000 | 0.000 | 0.236 |
| test set | 1.000 | 0.000 | 0.000 | 0.225 |

Table 8: The $p$-value of KS hypothesis test among MBF coefficients of different data sources.

| ImageNet-VGG-16 | clean | noisy | BIM | CW-L2 | DeepFool | R-PGD |
|---|---|---|---|---|---|---|
| (train set, test set) | 0.839 | 0.858 | 0.368 | 0.842 | 0.246 | 0.221 |
| (train set, out-of-sample set) | 0.049 | 0.054 | 0.094 | 0.407 | 0.737 | 0.088 |

whether two samples follow the same distribution (two-sample KS test). Specifically, in one-sample KS test, the distance between the empirical distribution function of one sample and the cumulative distribution function of the reference probability distribution is measured. Then, the $p$-value corresponding to the obtained distance is computed. If the $p$-value is larger than the significance level $\alpha$ (here we set $\alpha = 0.05$), then the null hypothesis that the sample follows the reference distribution is accepted; otherwise, rejected. Similarly, in two-sample KS test, the distance between the empirical distribution functions of two samples is computed. If the corresponding $p$-value is larger than $\alpha$, then it accepts that two samples follow the same distribution. The KS test conducted below is implemented by the python function $scipy.stats.ks\_2samp$[7].

**Hypothesis test 1**. Here we verify that whether the posterior vectors of both adversarial and benign examples follow GGD. We denote the posterior vector of one adversarial example as $\mathbf{p}_{\text{adv}}$, and that of one benign example as $\mathbf{p}_{\text{ben}}$. The distribution of GGD is denoted as $\mathcal{P}_{\text{GGD}}$. Then, we conduct two one-sample KS tests, including:

- **H1.1** The test of adversarial examples: $H_0$: $\mathbf{p}_{\text{adv}} \sim \mathcal{P}_{\text{GGD-adv}}$; $H_1$: $\mathbf{p}_{\text{adv}} \not\sim \mathcal{P}_{\text{GGD-adv}}$.
- **H1.2** The test of benign examples: $H_0$: $\mathbf{p}_{\text{ben}} \sim \mathcal{P}_{\text{GGD-ben}}$; $H_1$: $\mathbf{p}_{\text{ben}} \not\sim \mathcal{P}_{\text{GGD-ben}}$.

In **H1.1**, the reference distribution $\mathcal{P}_{\text{GGD-adv}}$ is firstly estimated from $\mathbf{p}_{\text{adv}}$, using the estimated method proposed in (Lasmar et al. (2009)). To alleviate the uncertainty of the estimation, we draw 500 samples from the estimated $\mathcal{P}_{\text{GGD-adv}}$. Then, we conduct the two-sample KS test between $\mathbf{p}_{\text{adv}}$ and these 500 samples respectively. The average $p$-value over 1000 tests is recorded. The mean of the average $p$-values over the whole database is reported. **H1.2** is conducted similarly. The results tested on ImageNet-AlexNet are shown in Table 6. Due to the space limit, results on other databases and models will be presented in the **supplementary material**. In all cases, the $p$-values are larger than the significance level $0.05$. Hence, we can conclude that *the posterior vectors of both adversarial and benign examples follow GGD*. However, note that the parameters of their corresponding GGD are different, which will be verified in the following test.

**Hypothesis test 2**. Here we verify that whether the extracted MBF features of adversarial and benign examples follow the same empirical distribution. We denote the MBF feature vector of one adversarial example as $\mathbf{m}_{\text{adv}}$, and the corresponding empirical distribution is denoted as $\hat{\mathcal{P}}_{\text{adv-MBF}}$. Similarly, we define $\mathbf{m}_{\text{ben}}$ and $\hat{\mathcal{P}}_{\text{ben-MBF}}$ for benign examples. Then, we conduct the following four two-sample KS tests:

- **H2.1** The test between adversarial and benign examples: $H_0$: $\hat{\mathcal{P}}_{\text{adv-MBF}} = \hat{\mathcal{P}}_{\text{ben-MBF}}$; $H_1$: $\hat{\mathcal{P}}_{\text{adv-MBF}} \neq \hat{\mathcal{P}}_{\text{ben-MBF}}$.
- **H2.2** The test between adversarial examples crafted from different attack methods: $H_0$: $\hat{\mathcal{P}}_{\text{adv-MBF}}^{\text{attack-1}} = \hat{\mathcal{P}}_{\text{adv-MBF}}^{\text{attack-2}}$; $H_1$: $\hat{\mathcal{P}}_{\text{adv-MBF}}^{\text{attack-1}} \neq \hat{\mathcal{P}}_{\text{adv-MBF}}^{\text{attack-2}}$.
- **H2.3** The test between adversarial examples from different data sources (*i.e.*, train, test and out-of-sample set): $H_0$: $\hat{\mathcal{P}}_{\text{adv-MBF}}^{\text{source-1}} = \hat{\mathcal{P}}_{\text{adv-MBF}}^{\text{source-2}}$; $H_1$: $\hat{\mathcal{P}}_{\text{adv-MBF}}^{\text{source-1}} \neq \hat{\mathcal{P}}_{\text{adv-MBF}}^{\text{source-2}}$.

---

[7]https://docs.scipy.org/doc/scipy-0.14.0/reference/generated/scipy.stats.ks_2samp.html

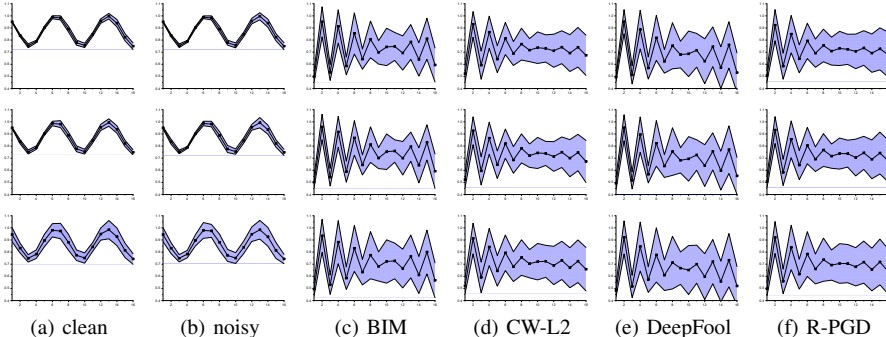

| (a) clean | (b) noisy | (c) BIM | (d) CW-L2 | (e) DeepFool | (f) R-PGD |

Figure 1: Statistics (mean $\pm$ standard deviation) of MBF coefficients on train (top row), test (median row), and out-of-sample (bottom row) set of ImageNet-VGG-16.

- **H2.4** The test between benign examples from different data sources (*i.e.*, train, test and out-of-sample set): $H_0$: $\hat{\mathcal{P}}_{\text{ben-MBF}}^{\text{source-1}} = \hat{\mathcal{P}}_{\text{ben-MBF}}^{\text{source-2}}$; $H_1$: $\hat{\mathcal{P}}_{\text{ben-MBF}}^{\text{source-1}} \neq \hat{\mathcal{P}}_{\text{ben-MBF}}^{\text{source-2}}$.

In above four two-sample KS tests, we test on the 16-dimensional MBF features extracted from the soft-max layer. Since the implementation $scipy.stats.ks\_2samp$ cannot compare two vectors, we compare the feature of each dimension separately, then report the average $p$-value over all dimensions. Specifically, when comparing two sets of samples, we firstly concatenate the feature of each dimension across all samples in the same set, leading to 16 long vectors for each set. Then, each pair of two long vectors corresponding to the same dimension from two sets is compared by KS test. The average $p$-value over all 16 dimensions is reported. The $p$-values of **H2.1** are shown in Table 7 (see the column "(clean, BIM)"). The $p$-values on both train and test set are 0. Thus, the hypothesis $H_0$ is rejected, *i.e.*, the MBF features of adversarial and benign examples follow different distributions. The $p$-values of **H2.2** are shown in Table 7. We pick two groups of attack methods, *i.e.*, (BIM, DeepFool) and (CW-L2, R-PGD). The $p$-values of (BIM, DeepFool) are 0, while the $p$-values of (CW-L2, R-PGD) are larger than $0.05$. It demonstrates that the distributions of adversarial examples crafted from different attack methods are possible to be different. The $p$-values of **H2.3** are shown in Table 8. The $p$-values of all types of adversarial examples exceed 0.05. Thus, the MBF features of adversarial examples from different data sources follow the same distribution. The $p$-values of **H2.4** are shown in Table 8. Only the $p$-value of "(train set, out-of-sample set)" of clean examples is slightly lower than 0.05, while the values of other cases exceed 0.05. Thus, in most cases, the MBF features of benign examples from different data sources follow the same distribution.

From above analysis, we obtain the following conclusions: **1)** The extracted MBF features of adversarial and benign examples follow different empirical distributions. It explains why MBF features are effective for detecting adversarial and benign examples; **2)** The extracted MBF features of adversarial/benign examples from the train, test and out-of-sample sets follow the same empirical distribution. Although the extracted MBF features of adversarial examples crafted from different attack methods may not follow the same empirical distribution, the significant difference between benign and different adversarial distributions can still lead to the good detection performance in the attack-transfer case. It explains why MBF features are robust across different attack methods and different data sources. Moreover, we visualize the statistics of each dimension of MBF features, *i.e.*, mean and standard deviation, as shown in Fig. 1. These visualizations also support above conclusions. Due to space limit, more KS tests and visualizations on different databases and networks will be presented in the **supplementary material**.

## 5 CONCLUSION

This work has proposed a novel adversarial detection method, dubbed MBF. The assumption behind is that the internal responses of the classification network of both adversarial and benign examples follow the generalized Gaussian distribution (GGD), but with different shape factors. The magnitude of Benford-Fourier coefficient is a function w.r.t. the shape factor, and can be easily estimated based on responses. Thus, it can serve as the discriminative features between adversarial and benign examples. The extensive experiments conducted on several databases, as well as the empirical analysis via KS test, demonstrate the superior effectiveness and robustness to different attacks and different data sources of the proposed MBF method, over state-of-the-art detection methods.

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

# A  APPENDIX

## A.1  PROOF OF THEOREM 1

**Proof 1** *Assume that all the variables $\{\mathcal{T}_m\}_{m=1,...,M}$ are independent and identically distributed. Applying the central limit theorem Rosenblatt (1956) to the real and imaginary parts of $\mathcal{Y}$, we can obtain that both parts asymptotically follow the Gaussian distribution*

$$\mathcal{Y} = \frac{1}{M}\sum_{m=1}^{M}\mathcal{T}_m \sim \mathcal{N}\big(E(\mathcal{T}_1), \frac{1}{M}D(\mathcal{T}_1)\big), \tag{10}$$

*where*

$$E(\mathcal{T}_1) = E\left(e^{-j2\pi n\log_{10}|\mathcal{X}_1|}\right) = \int_{-\infty}^{+\infty}\mathcal{P}_{\mathcal{X}_1}(x)\cdot e^{-j2\pi n\log_{10}|x|}dx = a_n, \tag{11}$$

$$D(\mathcal{T}_1) = E\big(|\mathcal{T}_1|^2\big) - |E(\mathcal{T}_1)|^2 = E\big(\left|e^{-j2\pi n\log_{10}|\mathcal{X}_1|}\right|^2\big) - |a_n|^2 = 1 - |a_n|^2. \tag{12}$$

*The PDF of $\mathcal{Y}$ can be rewritten as follows:*

$$\mathcal{Y} \sim \mathcal{N}(a_n, \frac{1-|a_n|^2}{M}). \tag{13}$$

*Thus, we obtain*

$$\mathcal{E} \sim \mathcal{N}(0, \frac{1-|a_n|^2}{M}). \tag{14}$$

*Besides, the pseudo variance Goodman (1963) of $\mathcal{T}_1$ is*

$$\mathcal{J}_{\mathcal{T}_1,\mathcal{T}_1} = E(\mathcal{T}_1^2) - E(\mathcal{T}_1)^2 = E\big(e^{-j2\pi n\log_{10}|\mathcal{X}_1^2|}\big) - a_n^2 = a_{2n} - a_n^2. \tag{15}$$

*Correspondingly,*

$$\mathcal{J}_{\mathcal{E},\mathcal{E}} = \frac{1}{M}\mathcal{J}_{\mathcal{T}_1,\mathcal{T}_1} = \frac{a_{2n}-a_n^2}{M}. \tag{16}$$

*Since $(a_{2n}-a_n^2)$ is bounded, we have $\lim_{M\to\infty}\mathcal{J}_{\mathcal{E},\mathcal{E}} = 0$. Thus, we obtain that the random variable $\mathcal{E}$ follows circularly-symmetric complex Gaussian distribution, because the sufficient and necessary condition is that mean value and pseudo variance equal zero Goodman (1963). It implies that both the real and imaginary part of $\mathcal{E}$ follow the same Gaussian distribution and they are independent. Thus, the magnitude of this complex random variable follows the Rayleigh distribution Papoulis & Pillai (2001), and the probability density function of $|\mathcal{E}|$ can be formulated as*

$$\mathcal{P}_{|\mathcal{E}|}(r) = \frac{r}{s^2}e^{-r^2/2s^2}, \tag{17}$$

*where $s$ is the scale parameter. Knowing the properties of the Rayleigh distribution Papoulis & Pillai (2001), we have:*

$$s^2 = \frac{1}{2}D(\mathcal{E}) = \frac{1-|a_n|^2}{2M}. \tag{18}$$

*Utilizing the fact that $|a_n|^2$ is close to 0 when n is a modest number, we obtain that $D(\mathcal{E}) \approx \frac{1}{M}$, leading to $s^2 = \frac{1}{2M}$. Then, we obtain*

$$E\big(|\mathcal{E}|\big) = \frac{1}{2}\sqrt{\frac{\pi}{M}}, D\big(|\mathcal{E}|\big) = \frac{4-\pi}{4M}. \tag{19}$$

*It is easy to observe that both $E\big(|\mathcal{E}|\big)$ and $D\big(|\mathcal{E}|\big)$ are close to zero when the number of samples M increases. It implies that the estimation error $\varepsilon_n$ gets closer to 0 as M increases.*

Table 9: The $p$-value of KS hypothesis test among clean samples, noisy samples, and adversarial samples crafted by four methods, respectively.

|  |  | clean | noisy | BIM | CW-L2 | DeepFool | R-PGD |
|---|---|---|---|---|---|---|---|
| CIFAR-10 | train set | 0.074 | 0.074 | 0.158 | 0.139 | 0.155 | 0.159 |
|  | test set | 0.072 | 0.072 | 0.158 | 0.138 | 0.216 | 0.216 |
| SVHN | train set | 0.104 | 0.104 | 0.249 | 0.194 | 0.222 | 0.245 |
|  | test set | 0.106 | 0.106 | 0.249 | 0.193 | 0.219 | 0.239 |
| ImageNet-AlexNet | train set | 0.116 | 0.116 | 0.253 | 0.235 | 0.242 | 0.253 |
|  | test set | 0.117 | 0.117 | 0.261 | 0.236 | 0.243 | 0.259 |
| ImageNet-VGG-16 | train set | 0.114 | 0.114 | 0.236 | 0.228 | 0.240 | 0.235 |
|  | test set | 0.114 | 0.114 | 0.231 | 0.228 | 0.239 | 0.232 |

Table 10: The $p$-value of two-sample KS test among MBF coefficients of different types of examples.

|  |  | (clean, noisy) | (clean, BIM) | (BIM, DeepFool) | (CW-L2, R-PGD) |
|---|---|---|---|---|---|
| CIFAR-10 | train set | 0.640 | 0.000 | 0.000 | 0.003 |
|  | test set | 0.764 | 0.000 | 0.030 | 0.000 |
| SVHN | train set | 0.559 | 0.000 | 0.030 | 0.000 |
|  | test set | 0.685 | 0.000 | 0.000 | 0.000 |
| ImageNet-AlexNet | train set | 1.000 | 0.000 | 0.002 | 0.134 |
|  | test set | 0.993 | 0.000 | 0.477 | 0.000 |
| ImageNet-VGG-16 | train set | 0.998 | 0.000 | 0.000 | 0.236 |
|  | test set | 1.000 | 0.000 | 0.000 | 0.225 |

Table 11: The $p$-value of KS test among MBF coefficients from different data sources.

|  |  | clean | noisy | BIM | CW-L2 | DeepFool | R-PGD |
|---|---|---|---|---|---|---|---|
| CIFAR-10 | (train set, test set) | 0.120 | 0.188 | 0.333 | 0.456 | 0.584 | 0.223 |
| SVHN | (train set, test set) | 0.494 | 0.708 | 0.523 | 0.147 | 0.491 | 0.645 |
| ImageNet-AlexNet | (train set, test set) | 0.153 | 0.148 | 0.370 | 0.605 | 0.410 | 0.386 |
|  | (train set, out-of-sample) | 0.000 | 0.000 | 0.169 | 0.101 | 0.206 | 0.158 |
| ImageNet-VGG-16 | (train set, test set) | 0.839 | 0.858 | 0.368 | 0.842 | 0.246 | 0.221 |
|  | (train set, out-of-sample) | 0.049 | 0.054 | 0.094 | 0.407 | 0.737 | 0.088 |

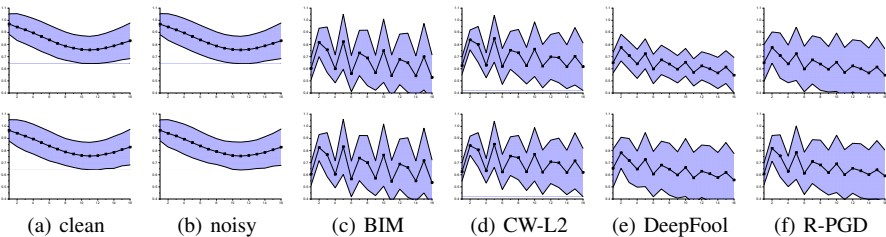

|  |  |  |  |  |  |
|---|---|---|---|---|---|
| (a) clean | (b) noisy | (c) BIM | (d) CW-L2 | (e) DeepFool | (f) R-PGD |

Figure 2: Statistics (mean $\pm$ standard deviation) of MBF coefficients on train (top row) and test (bottom row) set of CIFAR-10.

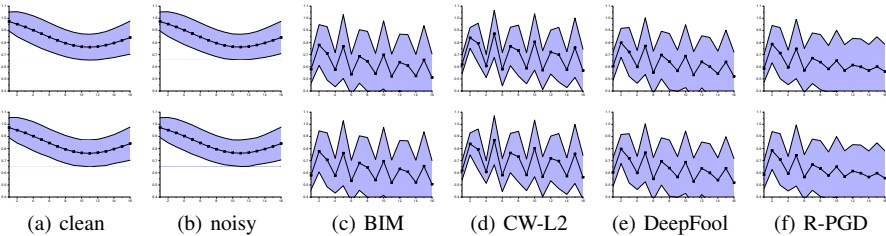

|  |  |  |  |  |  |
|---|---|---|---|---|---|
| (a) clean | (b) noisy | (c) BIM | (d) CW-L2 | (e) DeepFool | (f) R-PGD |

Figure 3: Statistics (mean $\pm$ standard deviation) of MBF coefficients on train (top row) and test (bottom row) set of SVHN.

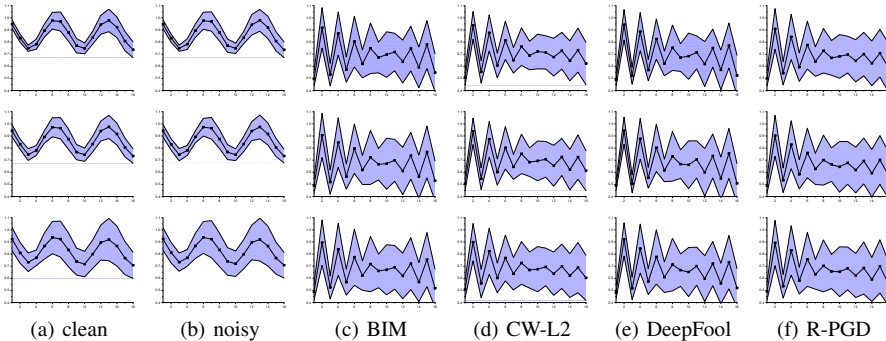

| (a) clean | (b) noisy | (c) BIM | (d) CW-L2 | (e) DeepFool | (f) R-PGD |

Figure 4: Statistics (mean ± standard deviation) of MBF coefficients on train (top row) and test (median row), and out-of-sample (bottom row) set of ImageNet-AlexNet.

Table 12: Detection results evaluated by AUROC score (%) on nearly 4000 images from the testset of MNIST. The best results are highlighted in bold.

| Iteration number in each GD run | Number of GD runs | BIM | CW-L2 | DeepFool | R-PGD |
|---|---|---|---|---|---|
| 200 | 10 | 0.744 | 0.980 | 0.702 | 0.746 |
| 100 | 10 | 0.689 | 0.973 | 0.650 | 0.690 |
| 100 | 5 | 0.664 | 0.969 | 0.628 | 0.666 |
| 100 | 2 | 0.612 | 0.959 | 0.586 | 0.614 |
| MBF | | **1.000** | **0.991** | **1.000** | **1.000** |

## A.2 ADDITIONAL EMPIRICAL ANALYSIS

Here we present additional empirical analysis on more databases and networks, as shown in Tables 9, 10 and 11. The $p$-values in most cases also supports the conclusions obtained in the main manuscript. We also present more visualizations in Figs. 2, 3 and 4. These visualizations also demonstrate the distinct difference of MBF features between adversarial and benign examples.

## A.3 COMPARISON BETWEEN DEFENSE-GAN

We adopt official code[8] of Defense-GAN (Samangouei et al. (2018)) and train an generative model on train set of MNIST dataset after $130,000$ epochs (As the official code of Defense-GAN currently only supports three datasets including MNIST, Fashion-MNIST, and CelebA). The classification model attains an accuracy of 98.7% on the test set of MNIST dataset. We still utilize four attack strategies, including BIM, CW-L2, DeepFool and R-PGD. After removing images that are misclassified and failed to attack, we pick 6000 images to train SVMs based on our MBF features, and leave nearly 4000 images for testing. There are two key hyper-parameters in Defense-GAN, including the iteration number in each GD run, and the total number of GD runs. We have tried multiple settings of these two hyper-parameters. Table 12 show the AUROC score on nearly $4000$ images. Our MBF method exceeds Defense-GAN by a large margin in quantitative performance.

## A.4 DETAILED PARAMETERS OF ADVERSARIAL ATTACK

Four popular adversarial attack methods are adopted to craft adversarial examples, including basic iterative method (BIM), CarliniWagnerL2Attack (CW-L2), DeepFool, and random projected gradient descent (R-PGD). We emphasize detailed parameters of these attack strategies for reproducibility, which are BIM (eps=0.3, stepsize=0.05, iterations=10), CW-L2 (binary_search_steps=5, confidence=0.0, learning_rate=0.005, max_iterations=1000), DeepFool (max_steps=100), R-PGD (eps=0.3, stepsize=0.01, iterations=40). All of the attack strategies are implemented by Foolbox.

---

[8] https://github.com/kabkabm/defensegan/

