# OpenReview forum: "Effective and Robust Detection of Adversarial Examples via Benford-Fourier Coefficients"
_ICLR.cc/2020/Conference — Reject_

### Official Review · AnonReviewer3 · 2019-10-21
**Official Blind Review #3**

**Rating:** 6

**Review:**

This paper proposes an adversarial detection method via Fourier coefficients. The proposed method seems promising, and empirical evaluations are reasonable.

However, I find that the proposed MBF detection metric is much more complicated to calculate than any of its baselines, e.g., LID or K-density. So I wonder if the good performance of MBF mainly comes from its ’complexity‘. I mean, if we use the feature vectors of different layers in CNNs and combine them with a different non-linear function and feed into an SVM classifier,  can we still obtain a hard-to-evade detector? I think a fair complexity is particularly important when you try to evade the detector by optimization-based adaptive attacks and claim superiority over other baselines.

**Experience Assessment:**

I have published in this field for several years.

**Review Assessment: Checking Correctness Of Derivations And Theory:**

I assessed the sensibility of the derivations and theory.

**Review Assessment: Checking Correctness Of Experiments:**

I carefully checked the experiments.

**Review Assessment: Thoroughness In Paper Reading:**

I read the paper thoroughly.

---

> ### Author Response · Authors · 2019-11-14
> **Response to Reviewer #3**
>
> Thanks for the constructive comments.
> 1)	The equation for computing the MBF feature, i.e., Eq. (8), may look “complicated”, since it involves sin and cosine function. But the truth is that it is rather simple, and the computational cost is only O(M_i^l) for the l-th layer of the i-th training image, with M_i^l being the number of entries in that layer. In contrast, the features computed in LID will utilize the responses of other training images, and its computation is more complicated than ours.
> 2)	However, the question that whether the complexity of features is important for an effective detector is very interesting. If we understand correctly, this sentence “if we use the feature vectors of different layers in CNNs and combine them with a different non-linear function and feed into an SVM classifier, can we still obtain a hard-to-evade detector” is asking that whether the discriminative features included in the responses could be approximated by one additional model (e.g., the neural network). The answer should be yes. As mentioned in the response to Reviewer 2 (see above), Defense-GAN actually do that. It is like to compare the hand-crafted features + shallow classifier (i.e., shallow learning) and the features learned together with the classifier (e.g., deep learning). Both our MBF method and LID belong to the same type, while Defense-GAN belongs to the second type. It is difficult to tell which type is better for adversarial detection. Both of them have advantages and limitations. The first type is computationally cheaper, as the features are computed following some fixed equations. In contrast, the second type has to train a new model, maybe for each database and each attack method, which is rather costly. Besides, the second type is likely to require much more training data than the first type to achieve satisfied performance, as there are much more parameters. At least on MNIST, as shown in the response to Reviewer 2, our MBF method performs much better than Defense-GAN. These discussions have been added into the revised manuscript.

---

### Official Review · AnonReviewer2 · 2019-10-25
**Official Blind Review #2**

**Rating:** 6

**Review:**

This paper proposes an approach to adversarial detection.  The approach first computes a representation of the activation layers using the Benford-Fourier coefficients.  One then generates a range of noisy instances, and trains an SVM using those noisy instances as supervised labels (e.g., noisy instances are adversarial).  The SVM uses the Benford-Fourier coefficients of the activation layer as the input features.  The results show good performance against some baselines such as LID.

I'm not really an expert in this area, but I'm a bit surprised that LID is considered the baseline to beat.  I imagine that most adversarial defense approaches are for robust prediction, rather than detection.  It also seems the authors chose to compare with defenses that are computationally cheaper (so not RCE or Defense-GAN), but a study of computational trade-offs is absent in the paper.

**Experience Assessment:**

I have read many papers in this area.

**Review Assessment: Checking Correctness Of Derivations And Theory:**

I assessed the sensibility of the derivations and theory.

**Review Assessment: Checking Correctness Of Experiments:**

I assessed the sensibility of the experiments.

**Review Assessment: Thoroughness In Paper Reading:**

I read the paper at least twice and used my best judgement in assessing the paper.

---

> ### Author Response · Authors · 2019-11-14
> **Response to Reviewer #2**
>
> Thanks for the constructive comments.
> 1)	We should clarify that the reason we choose LID as the baseline is not the cheap computation. The real reason is that LID can be seen one state-of-the-art adversarial detection methods, as it has been cited by 114 times since its publication at ICLR 2018, and it is compared in many recent works about adversarial detection. Our experimental results also show that LID is the most competitive among all compared methods. Another important reason is that LID and our MBF method share the similar idea that extracting discriminative features for detection from the responses, based on some assumptions. LID assumes that the features of adversarial and benign have different densities, while our MBF method assumes that they follow different GGDs.
> 2)	In contrast, Defense-GAN is designed using a totally different philosophy with LID and our MBF method, i.e., approximating the adversarial noises using GAN. Since the official code of Defense-GAN only supports the three datasets of MNIST, Fashion-MNIST, and CelebA, we compare with Defense-GAN on MNIST, as shown below. There are two key hyper-parameters in Defense-GAN, including the iteration number in each GD run, and the total number of GD runs. After trying multiple settings of these two hyper-parameters, we report the best result (evaluated by AUROC score). Our MBF method show much better performance than Defense-GAN. The detailed comparisons with Defense-GAN have been added in the revised manuscript.
> __________________________________________________________________________________________________________
>  Iteration number in each GD run | Number of GD runs |  BIM  | CW-L2 | DeepFool | R-PGD
> __________________________________________________________________________________________________________
>                              200                          |                 10                | 0.744 |  0.980  |     0.702     |  0.746
> __________________________________________________________________________________________________________
>                                                            MBF                                  | 1.000 |  0.991  |     1.000     |  1.000
> __________________________________________________________________________________________________________

---

### Official Review · AnonReviewer1 · 2019-10-28
**Official Blind Review #1**

**Rating:** 3

**Review:**

This paper presents a new discriminator metric for adversarial attack's detection by deriving the different properties of l-th neuron network layer on different adv/benign samples. This method can achieve good AUC score comparing to other start-of-art detection methods and also achieve good robustness under corresponding adaptive attack. The framework is clear and the experiment is solid.

However, I have several concerns:

Major:
1. It seems that the whole process assumes that there is difference for the parameters in the environment of GGD with adv/benign samples, and the goal is to search for the major components of it and use a classifier to detect. To extract the approximation of parameters,  the authors use the "response entries" of l-th layer for several observations => this means the authors regard all the "response entries" of one layer as different samples on one certain GGD. This makes me feel a bit tricky, and it would be great if you the authors can provide some evidence or explanation here.

2. In the experiment's remark, the authors mentioned that the mean parameter of GGD is set to 0 and most of them are actually close to 0 (around 1e-2) so the assumption is right. However 1e-2 is not a value "very close to zero" and it would be great to show / explain the variance here.

3. I can't find the parameter of your evaluated attack method (like confidence, eps, etc.) Please also provide experimental details for reproducibility.

Minor:

Here are some reference error (e.g. P6, "For each database, as described in Section ??"). Please fix that.

Overall, this paper is a interesting based on the performance of detection. But the  assumptions made by the paper are a bit confusing and it would be good to clarify and provide clarification for them. Authors should explain the assumptions and give some extra experiment results if needed.


**Experience Assessment:**

I have published in this field for several years.

**Review Assessment: Checking Correctness Of Derivations And Theory:**

I carefully checked the derivations and theory.

**Review Assessment: Checking Correctness Of Experiments:**

I carefully checked the experiments.

**Review Assessment: Thoroughness In Paper Reading:**

I read the paper thoroughly.

---

> ### Author Response · Authors · 2019-11-14
> **Response to Reviewer #1**
>
> We thank the reviewer for the comments and appreciation, and would like to answer the reviewer’s questions as follows:
> Major 1.
> Yes, we indeed assume that “all the response entries of one layer as different samples on one certain GGD”. However, it should be emphasized that, as demonstrated above Eq. (5), “assume that x = {x1, …, x_M} is a set of M i.i.d. points sampled from GGD with the same shape factor”. It means that we only assume the shape factor of GGD is same, while the parameters mean and standard deviation could be different. Thus, we only extract the features of the magnitude of Benford-Fourier coefficient which could estimate the shape factor as the discriminative features between adversarial and benign examples.
> To validate this assumption, we present hypothesis tests for both benign and adversarial examples, in Section 4.5. As shown in Table 6, the p-values under different attacks are all over 5%, that means the hypothesis H0 is accepted, i.e., the benign/adversarial responses follow GGD with the same shape factor. Thus, we can claim that this assumption is statistically true. Our experimental results also support this assumption.
> Finally, we try to clarify the rationale behind this assumption once again. 1) We believe the features (including the internal responses and the fully-connected features) of adversarial examples should also follow some distributions, similar to benign examples. If not that case, it is difficult to understand/explain why the adversarial examples (generated by different attack methods and on different benign examples) could be enforced to avoid the original class (untargeted attack) or to be predicted as one certain target class (targeted attack). 2) We don’t believe that the features of adversarial and benign examples simply follow different Gaussian distributions, as the parameter mean and standard deviation of Gaussian distribution could vary significantly among different data sources and attack methods. These adversarial examples should be similar to each other and different to benign examples at some higher-level perspectives. Thus, we choose the GGD distribution, which is very general to cover many widely used distributions. And, the shape factor of GGD is more robust than the mean and standard deviation, across different data sources. This is why we propose such an assumption.
> Hope above explanations could provide some informative messages to understand the proposed assumption.
>
> Major 2.
> Thanks for this helpful suggestion. We do a detailed statistics of all response layers on CIFAR-10. We find that although the mean parameters at most layers are close to 0, the mean parameters at a few layers could be large. Specifically, for each image and each layer, we compute one mean parameter. Then, for each layer, we compute the average and standard deviation of the mean parameters across the whole database. The average values of all layers range from 6e-3 to 6.9, and the standard deviation of all layers range from 0 to 4.3. However, in our experiments, the mean value is subtracted from each response entry when we extract the MBF features. Thus, the assumption that the mean parameter of GGD is always satisfied. This point has been clarified in the revised manuscript.
>
> Major 3.
> Thanks for this helpful suggestion. In experiments, we adopt the default parameter setting in Foolbox (v1.8.0). Specifically, the default parameters of each attack methods are:
> ‘BIM’(eps=0.3, stepsize=0.05, iterations=10); ‘CarliniWagnerL2Attack’(binary_search_steps=5, confidence=0, learning_rate=0.005, max_iterations=1000); ‘DeepFoolAttack’(max_steps=100); ‘RandomPGD’(epsilon=0.3, stepsize=0.01, iterations=40). These have been clarified in the revised manuscript.
>
> Minor 1.
> Thanks. It has been corrected in the revised manuscript.

---

### Decision · Program_Chairs · 2019-12-19

**Decision:**

Reject

**Comment:**

This paper presents a new metric for adversarial attack's detection. The reviewers find the idea interesting, but the some part has not been clearly explained, and there are questions on the reproducibility issue of the experiments.